# Dynamical control of quantum heat engines using exceptional points

J.-W. Zhang[1,2,10], J.-Q. Zhang[1,10], G.-Y. Ding[1,3,10], J.-C. Li[1,3], J.-T. Bu[1,3], B. Wang[1,3], L.-L. Yan[4], S.-L. Su[4], L. Chen[1,2], F. Nori[5,6], Ş. K. Özdemir[7], F. Zhou[1,2] ✉, H. Jing[8,9] ✉ & M. Feng[1,2,4] ✉

A quantum thermal machine is an open quantum system coupled to hot and cold thermal baths. Thus, its dynamics can be well understood using the concepts and tools from non-Hermitian quantum systems. A hallmark of non-Hermiticity is the existence of exceptional points where the eigenvalues of a non-Hermitian Hamiltonian or a Liouvillian superoperator and their associated eigenvectors coalesce. Here, we report the experimental realization of a single-ion heat engine and demonstrate the effect of Liouvillian exceptional points on the dynamics and the performance of a quantum heat engine. Our experiments have revealed that operating the engine in the exact- and broken-phases, separated by a Liouvillian exceptional point, respectively during the isochoric heating and cooling strokes of an Otto cycle produces more work and output power and achieves higher efficiency than executing the Otto cycle completely in the exact phase where the system has an oscillatory dynamics and higher coherence. This result opens interesting possibilities for the control of quantum heat engines and will be of interest to other research areas that are concerned with the role of coherence and exceptional points in quantum processes and in work extraction by thermal machines.

Quantum heat engines extract useful work from thermal reservoirs using quantum matter as their working substance. Contrary to their classical counterparts, which do not include coherence in its microscopic degrees of freedom and suffer from irreversible loss during a classical thermodynamic cycle, quantum heat engines are expected to benefit from quantum features to surpass the output power and efficiency that can be attained by an equivalent classical heat engine[1–4]. The growing interest in quantum heat engines is also fueled by the interest in understanding the quantum-classical transition in energy-information and work-heat conversion. Additional motivations include the need to maximize the efficiency (the ratio of useful work to the input heat) and the output power while keeping power fluctuations minimal in micro- and nano-scale heat engines, in which quantum fluctuations and non-equilibrium dynamics play a crucial role[5–9]. Microscopic and nanoscopic heat engines with and without the involvement of quantum coherences have been implemented with

[1]State Key Laboratory of Magnetic Resonance and Atomic and Molecular Physics, Wuhan Institute of Physics and Mathematics, Innovation Academy of Precision Measurement Science and Technology, Chinese Academy of Sciences, Wuhan, China. [2]Research Center for Quantum Precision Measurement, Guangzhou Institute of Industry Technology, 511458 Guangzhou, China. [3]School of Physics, University of the Chinese Academy of Sciences, 100049 Beijing, China. [4]School of Physics, Zhengzhou University, 450001 Zhengzhou, China. [5]Theoretical Quantum Physics Laboratory, RIKEN, Cluster for Pioneering Research, Wako-shi, Saitama 351-0198, Japan. [6]Physics Department, The University of Michigan, Ann Arbor, MI 48109-1040, USA. [7]Department of Engineering Science and Mechanics, and Materials Research Institute, Pennsylvania State University, State College, University Park, PA 16802, USA. [8]Key Laboratory of Low-Dimensional Quantum Structures and Quantum Control of Ministry of Education, Department of Physics and Synergetic Innovation Center for Quantum Effects and Applications, Hunan Normal University, 410081 Changsha, China. [9]Synergetic Innovation Academy for Quantum Science and Technology, Zhengzhou University of Light Industry, 450002 Zhengzhou, China. [10]These authors contributed equally: J.-W. Zhang, J.-Q. Zhang, G.-Y. Ding. ✉e-mail: zhoufei@wipm.ac.cn; jinghui73@foxmail.com; mangfeng@wipm.ac.cn

single trapped ions[8–10], ensembles of nitrogen-vacancy centres in diamond[4], magnetic resonance[11,12], a single electron box[13], and impurity electron spins in a silicon tunnel field-effect transistor[14]. Many interesting proposals have also been put forward for their realization in superconducting circuits[15,16] and optomechanics[17,18].

Another field that has been attracting increasing interest is non-Hermiticity, including parity-time ($\mathcal{PT}$) symmetry [19] in physical systems. In particular, non-Hermitian spectral degeneracies known as exceptional points (EPs) have been shown to have tremendous effects on the dynamics of physical systems, leading to many counterintuitive features which have led to the development of novel functionalities and classical devices [20–27]. Effects of non-Hermiticity have generally been studied using an effective non-Hermitian Hamiltonian and its spectral degeneracies. Recently, there is a growing interest to harness non-Hermiticity and EPs for quantum applications[28–33]. In quantum systems, however, Hamiltonian EPs (known as HEPs) cannot capture the whole dynamics because these exclude quantum jumps and the associated noises. Instead, one should resort to the Liouvillian formalism which takes into account both coherent non-unitary evolution and quantum jumps[34–38]. In this formalism, EPs are defined as the degenerate eigenvalues of Liouvillian superoperators. Thus they are referred as Liouvillian EPs (LEPs), whose properties and effects on quantum systems have remained largely unexplored except in some recent experiments in superconducting qubit systems [38,39].

As open quantum systems which exchange energy with thermal reservoirs, quantum heat engines naturally exhibit non-Hermitian dynamics, which can be controlled by judiciously tuning parameters of the heat engine to operate it in the exact- or broken-phases separated by LEPs (i.e., LEPs correspond to the transition points between the exact- and the broken- phases). Here we report the experimental implementation of a quantum Otto engine using a single $^{40}$Ca$^{+}$ ion confined in a linear Paul trap[40–42], and demonstrate the control of the engine efficiency and output power by harnessing LEPs and their associated dynamics. This constitutes an interesting observation of the signatures of LEPs in a quantum heat engine. We note that previous experiments studied non-Hermiticity in single-spin systems by considering only HEPs[28,31,32]. In contrast, here we use LEPs and their ramifications to control the performance of a quantum heat engine. Thus, our study takes into account quantum jumps and the associated dynamics. As it will become clear below and discussed previously[37,38], the LEP in this system corresponds to the critical damping point which emerges in the parameter space as the system transits between the oscillatory and non-oscillatory dynamics, in analogy with a damped harmonic oscillator.

## Results

### Single-ion quantum heat machine

The working substance of the quantum Otto engine we implement here is a pseudo-spin 1/2 represented by the internal states of a trapped single $^{40}$Ca$^{+}$ ion, i.e., the ground state $|4^2S_{1/2}, m_J = +1/2\rangle$ labeled as $|g\rangle$, and the metastable state $|3^2D_{5/2}, m_J = +5/2\rangle$ labeled as $|e\rangle$, with the magnetic quantum number $m_J$ (see Fig. 1A). In our experiment, we confine a single $^{40}$Ca$^{+}$ ion in a linear Paul trap, whose axial and radial frequencies are $\omega_z/2\pi = 1.1$ MHz and $\omega_r/2\pi = 1.6$ MHz, respectively. We define a quantization axis along the axial direction by a magnetic field of ~3.4 Gauss at the center of the trap. We then perform Doppler and sideband cooling of the ion until an average phonon number of $\bar{n} < 1$ with the Lamb-Dicke parameter ~0.11 is achieved. This is sufficient to avoid thermal phonons yielding offsets of Rabi oscillations[40–42], and thus observe quantum effects [42]. Populations of different energy levels of the qubit are detected using a photomultiplier tube (PMT) that monitors the fluorescence due to spontaneous decay from the excited state[41]. In our experiment, we observe the variation of the population in $|e\rangle$, from which the required thermodynamic quantities, such as work, output power and

efficiency, can be acquired. The temperature $T$ and the entropy $S$ of a spin system are defined as $T = -\Delta_0[k_B \ln(P_e/P_g)]^{-1}$ and $S = -k_B \text{Tr}[\rho \ln \rho]$, where $P_e$ and $P_g$ denote the populations of the states $|e\rangle$ and $|g\rangle$, respectively, $\Delta_0 = E_e - E_g$ is the effective energy gap between the states $|e\rangle$ and $|g\rangle$ in the interaction representation, $k_B$ is the Boltzmann constant, and $\rho$ is the density operator describing the state of the system.

Under proper laser irradiation as in Fig. 1B, we obtain an effective two-level model with engineered drive and decay (Supplementary Note 1), as shown in Fig. 1C, which can be described by the Lindblad master equation,

$$\dot{\rho} = -i[H,\rho] + \frac{\gamma_{\text{eff}}}{2}(2\sigma_-\rho\sigma_+ - \sigma_+\sigma_-\rho - \rho\sigma_+\sigma_-) \equiv \mathcal{L}\rho, \quad (1)$$

where $\mathcal{L}$ is the Liouvillian superoperator, $\rho$ denotes the density operator, $\gamma_{\text{eff}}$ is the effective decay rate from the excited state $|e\rangle$ to the ground state $|g\rangle$ (Supplementary Note 1), and

$$H = \Delta|e\rangle\langle e| + \frac{1}{2}\Omega(t)|e\rangle\langle g| + \text{H.c.}, \quad (2)$$

with $\Delta$ denoting the frequency detuning between the driving laser and the resonance transition of the ion and $\Omega(t)$ representing the drive amplitude (i.e., Rabi frequency) of the laser[40–42]. The dynamics of the system is fully captured by the Liouvillian superoperator $\mathcal{L}$ whose eigenvalues for $\Delta = 0$ are $\lambda_1 = 0$, $\lambda_2 = -\gamma_{\text{eff}}$, $\lambda_3 = (-3\gamma_{\text{eff}} - \xi)/4$, and $\lambda_4 = (-3\gamma_{\text{eff}} + \xi)/4$, with $\xi = \sqrt{\gamma_{\text{eff}}^2 - 16\Omega^2}$ (Supplementary Note 2 and Supplementary Fig. 1). When $\xi = 0$, that is $\gamma_{\text{eff}} = 4\Omega$, the eigenvalues $\lambda_3$ and $\lambda_4$ merge, giving rise to a second order LEP at $\tilde{\lambda} = -3\gamma_{\text{eff}}/4$. Clearly, for $\gamma_{\text{eff}} > 4\Omega$ (weak coupling), both $\lambda_3$ and $\lambda_4$ are real with a splitting amount $\xi$, corresponding to the broken phase characterized by a non-oscillatory dynamics with purely exponential decay[43,44]. For $\gamma_{\text{eff}} < 4\Omega$ (strong coupling), on the other hand, $\lambda_3$ and $\lambda_4$ form a complex conjugate pair which splits in their imaginary parts by $\xi$, corresponding to the exact phase characterized by an oscillatory dynamics. Thus, the LEP divides the parameter space into a region of oscillatory dynamics (exact phase, $\gamma_{\text{eff}} < 4\Omega$) and a region of non-oscillatory dynamics (broken phase, $\gamma_{\text{eff}} > 4\Omega$). As such, the LEP here is similar to the critical damping point of a damped harmonic oscillator and emerges in the transition between the regions of oscillatory (exact phase) and non-oscillatory (broken phase) dynamics.

### Quantum Otto cycles of a single ion

The question we address in this study is: How do the presence of LEPs and the associated transitions between the oscillatory and non-oscillatory dynamics affect the performance of an Otto engine? A typical Otto cycle has four strokes: two adiabatic strokes, which result in compression and expansion, and two isochoric strokes which connect the working substance to cold and hot baths. Quantum Otto cycles differ from their classical counterparts in the varying and the invariant thermodynamic quantities, and how these quantities are defined, see Supplementary Note 3 and Supplementary Fig. 2. For example, in a quantum isochoric stroke, the population $P_n$ of each level $n$ of the qubit, and hence the entropy $S$ of the system, changes until the working substance reaches thermal equilibrium with the heat bath, while there is no change in the eigenenergies $E_n$[14,45,46].

In a classical isochoric process, the pressure $P$ and the temperature $T$ change but the volume $V$ remains unchanged, and the working substance reaches thermal equilibrium with the heat bath only at the end of this process. In a classical adiabatic stroke, all thermodynamic quantities $P$, $T$, and the volume $V$ vary (i.e., no invariant thermodynamic quantity) and there is no requirement that occupation probabilities remain unchanged. Therefore, work is done only during

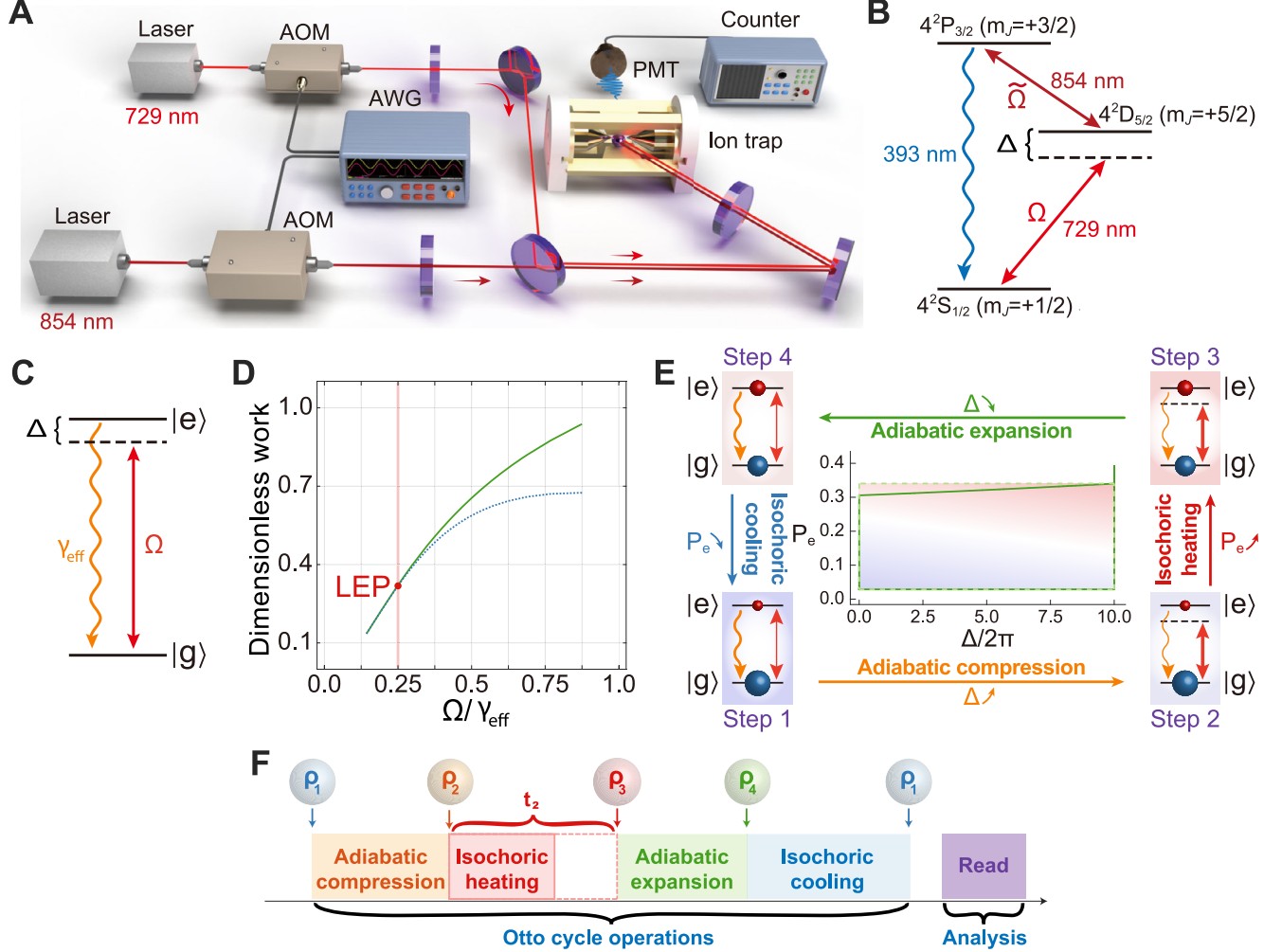

**Fig. 1 | Single-spin quantum heat engine in a trapped $^{40}$Ca$^+$ ion exhibiting a Liouvillian exceptional point (LEP). A** Schematic of the experimental setup. AOM: acousto-optic modulator. PMT: photomultiplier tube. AWG: arbitrary waveform generator. **B** Energy levels of the $^{40}$Ca$^+$ ion, where the straight red arrows represent transitions by laser irradiation with wavelengths labeled and the blue wavy arrow denotes spontaneous emission. Such a three-level configuration equals an effective two-level system with controllable driving and decay, as plotted in (**C**). **D** Schematic diagram for work done in the exact- and broken-phases separated in the parameter space by an LEP at $\Omega/\gamma_{\rm eff}=1/4$, where a bifurcation occurs due to coherence-induced oscillations in the exact phase. **E** Four strokes of our quantum Otto engine, where strokes from Step 1 to Step 2 and from Step 3 to Step 4 are adiabatic processes; while strokes from Step 2 to Step 3 and from Step 4 to Step 1 represent isochoric processes. The green dashed line represents an ideal quantum Otto cycle, and the solid line corresponds to the cycle obtained by solving the master equation using experimentally available parameter values. **F** Experimental operation sequences for an Otto cycle, where the duration $t_2$ of the second stroke (isochoric heating) is varied to quantify the quantumness involved in the cycle.

the classical adiabatic strokes (no work is done during classical isochoric strokes). Similarly, a quantum heat engine does work only during the quantum adiabatic strokes (i.e., no work is done during the quantum isochoric strokes) but with a different underlying mechanicsm than the classical adiabatic strokes: In a quantum adiabatic stroke, $P_n$ and $S$ should remain unchanged during the process (thus no heat exchange) but $E_n$ may shift. This change in $E_n$ leads to non-zero work.

As demonstrated later, we find that the coherence-enabled improvements in work and power output of a quantum heat engine arise if the coherence during the work strokes (i.e., quantum adiabatic strokes) induces a hump in the thermal strokes (i.e., quantum isochoric strokes). Therefore, to answer the question stated above and clarify the relation among LEPs, the surviving coherence after the thermal strokes, and the performance of a quantum heat engine, we have designed experiments implementing Otto cycles with (i) both isochoric strokes in the exact phase ($\gamma_{\rm eff} < 4\Omega$, oscillatory dynamics), (ii) both isochoric strokes in the broken phase ($\gamma_{\rm eff} > 4\Omega$, non-oscillatory dynamics), and (iii) isochoric heating stroke in the exact but isochoric cooling stroke in the broken phases. We note that a

fourth case would be the opposite of (iii), corresponding to a quantum refrigerator, totally reversing the process of the heat engine under the setting (iii). This case is beyond the scope of the present study and thus we have not performed experiments under this setting.

In our system of a single trapped ion, we implement the ingredients of the Otto cycle as follows: Hot and cold heat baths are prepared by tuning $\Omega/\gamma_{\rm eff}$, which is the ratio of the driving laser beam strength $\Omega$ to the effective decay $\gamma_{\rm eff}$ of the qubit. This implies that laser irradiation together with the real environment constitutes the baths, where the hot and cold baths correspond to strong and weak drives, respectively. The qubit absorbs or releases heat by coupling to the hot or the cold baths, respectively. Here, we adjust $\gamma_{\rm eff}$ by varying the power of the laser with wavelength 854 nm, which is tuned to the $P_{3/2}$-$D_{5/2}$ transition and $\Omega$ is adjusted by tuning the power of the 729 nm laser red-tuned to the $S_{1/2}$-$D_{5/2}$ transition (Fig. 1B). We evaluate the performance of the heat engine, e.g., the work output to the cold bath and heat absorbed from the hot bath, by monitoring the variation in the populations of the two-level system.

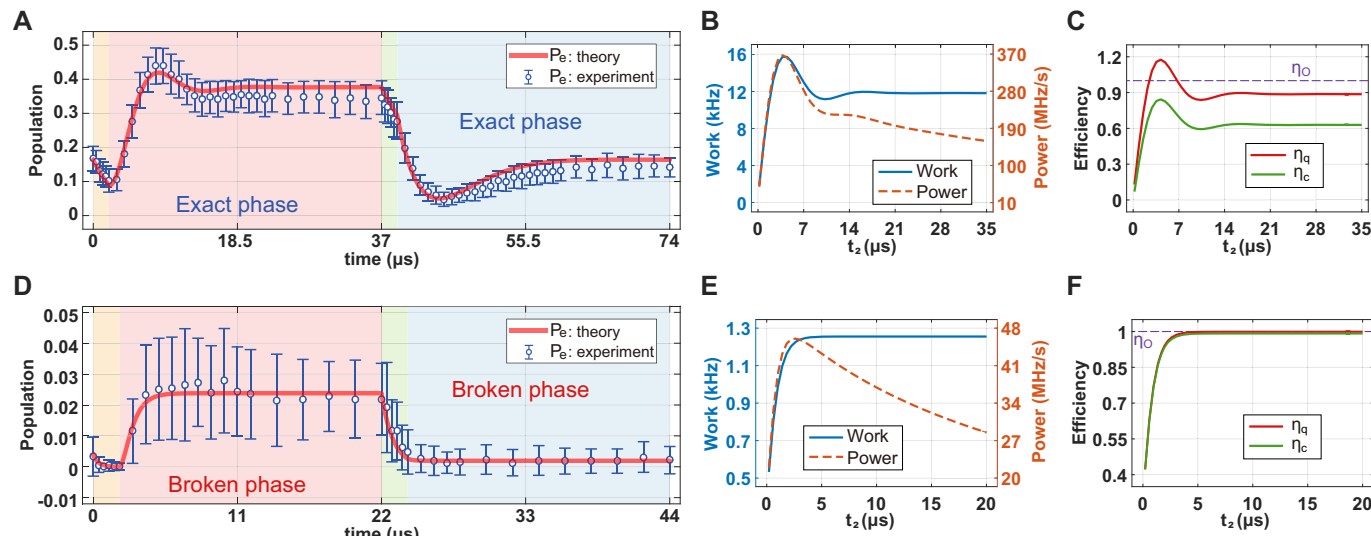

**Fig. 2 | Otto cycles with the isochoric processes executed in the exact- or the broken-phase. A** and **D** Time evolution of the population in the excited state $|e\rangle$ when both the second (isochoric heating) and the fourth (isochoric cooling) strokes are in the exact (strong drive and oscillatory dynamics)- and broken (weak drive and non-oscillatory dynamics) phases, respectively. The red solid curves are obtained by simulating the master equation. The circles and the error bars, respectively, denote the average and standard deviation of 10,000 measurements. Regions with different colors correspond to different strokes of the Otto cycle, with the orange, pink, green, and blue corresponding to the first (adiabatic compression), second (isochoric heating), third (adiabatic expansion), and fourth (isochoric cooling) strokes, respectively. The first (orange) and the third (green) strokes are implemented by up- and down-scanning the detuning $\Delta$ between $\Delta_{min}/2\pi = 0$ kHz, and $\Delta_{max}/2\pi = 10$ kHz,

respectively, with constant $\Omega$ and $\gamma_{eff}$. The second (pink) and the fourth (blue) strokes are implemented by rapidly increasing and decreasing $\Omega/\gamma_{eff}$ with constant detuning. In (**A**), we used $\{\Omega/2\pi = 23$ kHz, $\gamma_{eff} = 300$ kHz$\}$, and $\{\Omega/2\pi = 24$ kHz, $\gamma_{eff} = 120$ kHz$\}$ for the first and third strokes, respectively, and $\{\Omega/2\pi = 82$ kHz, $\gamma_{eff} = 370$ kHz, $\Delta = \Delta_{max}\}$ and $\{\Omega/2\pi = 24$ kHz, $\gamma_{eff} = 299$ kHz, $\Delta = \Delta_{min}\}$ for the second and fourth strokes. In (**D**), we used $\{\Omega/2\pi = 25$ kHz, $\gamma_{eff} = 2500$ kHz$\}$ and $\{\Omega/2\pi = 25$ kHz, $\gamma_{eff} = 970$ kHz$\}$ for the first and third strokes, respectively, and $\{\Omega/2\pi = 64$ kHz, $\gamma_{eff} = 2500$ kHz, $\Delta = \Delta_{max}\}$ and $\{\Omega/2\pi = 25$ kHz, $\gamma_{eff} = 2500$ kHz, $\Delta = \Delta_{min}\}$ for the second and fourth strokes. **B** and **E** The net work and output power, and (**C**) and (**F**), the efficiencies $\eta_c$ and $\eta_q$ as functions of the second stroke execution time $t_2$ while the execution times of the other strokes are kept fixed. In (**C**) and (**F**), the horizontal dashed lines represent the efficiency $\eta_O = 1$ of the ideal Otto cycle.

In our treatment under the rotating frame with respect to the driving laser frequency, quantum adiabatic strokes are executed by tuning the frequency of the driving laser which helps vary the internal energy gap $\Delta$ (but without population change) and the temperature of the working substance. Similarly, quantum isochoric strokes are performed by tuning $\Omega/\gamma_{eff}$ which controls the heat exchange between the working substance and the thermal baths. Thus, while the power of the 854 nm laser helps tune the qubit decay, the power and the frequency of the 729 nm laser helps implement the four strokes of the Otto cycle as follows (Fig. 1E, F and Supplementary Fig. 2): Starting with the qubit at steady state with a small population in the excited state, we first carry out an adiabatic compression by increasing the detuning $\Delta$ linearly from $\Delta_{min}$ to $\Delta_{max}$, where $P_e$ remains in a small and constant value (Step 1 → Step 2). Next, we perform isochoric heating by rapidly increasing $\Omega/\gamma_{eff}$ to a large value, during which the detuning remains equal to $\Delta_{max}$ (Step 2 → Step 3). Then, we carry out an adiabatic expansion by linearly decreasing $\Delta$ from $\Delta_{max}$ to $\Delta_{min}$, with $P_e$ staying unchanged (Step 3 → Step 4). Finally, we perform isochoric cooling by rapidly decreasing $\Omega/\gamma_{eff}$ to a small value with the $\Delta$ remaining unchanged as $\Delta_{min}$ (Step 4 → Step 1). To accomplish a closed Otto cycle, we wait, after finishing the last stroke, for the system reaching the steady state and returning to the initial state.

**Performances of the single-ion quantum heat engine**
We evaluate the role of coherence in the performance of the quantum heat engine by monitoring the oscillations in the populations of the qubit states in the isochoric stroke and then assess the net work, the output power, and the efficiency of the heat engine as a function of the execution time $t_2$ of the second stroke (isochoric heating) while the execution times of other strokes are kept fixed (Supplementary Notes 4 and 5).

The strong coupling and the exact-phase regimes overlap for $\Omega/\gamma_{eff} > 1/4$ and similarly the weak coupling and the broken-phase regimes overlap for $\Omega/\gamma_{eff} < 1/4$, with the LEP at $\Omega/\gamma_{eff} = 1/4$ being the transition point. Our theoretical study and numerical simulations indicate that in the broken phase (weak coupling regime), the net work produced in an Otto cycle linearly increases with increasing $\Omega/\gamma_{eff}$. As $\Omega/\gamma_{eff}$ is increased to transit into the exact phase (strong coupling), the net work increases with $\Omega/\gamma_{eff}$ in a slower and bifurcated fashion due to enhancement of the coherence in the isochoric strokes, as shown in Fig. 1D. Therefore, as we will show below, by tuning $\Omega/\gamma_{eff}$, and hence the competition between the driving field strength and the qubit decay, one may elucidate the effects of LEPs, non-Hermiticity, and the associated changes in coherence on the net work, output power, and efficiency of the quantum heat engine.

First, we implement both of the isochoric strokes in the exact phase (i.e., the regime with complex conjugate eigenvalue pairs and hence with oscillatory dynamics). The time evolution of the excited state population $P_e$ during the full Otto cycle exhibits a hump in the second stroke (i.e., isochoric heating) and a ramp in the fourth stroke (i.e., isochoric cooling), which reflect the sufficient coherence involved due to the strong coupling effect (Supplementary Note 4 and Supplementary Fig. 4), in full agreement with the results obtained from the simulations of the master equation (Fig. 2A). These features are also seen when the output power and the net amount of work are obtained as a function of the execution time $t_2$ of the second stroke, while the execution time of the other strokes are kept fixed (Fig. 2B). The clear hump at $t_2 \approx 4\,\mu$s is an indication of the role of coherence in the second stroke on the output power and the net work. We evaluate the efficiency of the Otto cycles in our experiments by calculating both the conventional efficiency $\eta_c$ (defined as the ratio of the net work to the heat absorbed during a full Otto cycle) and the heat absorption efficiency $\eta_q$ of the second stroke (i.e., quantum engine efficiency), which

corresponds to the energy difference between the initial and end time points of the second stroke (Supplementary Note 3). We find that (Fig. 2C): (i) both $\eta_c$ and $\eta_q$ exhibit humps with a trend similar to that observed in the net work and output power plots, implying the involvement of coherence in the process, (ii) $\eta_c$ is less than the efficiency $\eta_O = 1$ of the ideal Otto cycle, that is $\eta_c < \eta_O$, and (iii) $\eta_c < \eta_q$, indicating heat absorption during both the isochoric heating and the isochoric cooling strokes (2nd and 4th strokes).

Heat absorption during the isochoric cooling stroke is expected only in quantum systems and is a result of coherence (i.e., see the ramp in Fig. 2A). This additional heat absorption in the cooling stroke might be the reason for $\eta_c < \eta_O$, which suggests that for improved $\eta_c$, one should prevent heat absorption during the cooling stroke by minimizing, if not removing, the coherence during this cycle. We note that there is a slight reduction in the population of the excited state $|e\rangle$ during the first and third strokes (Fig. 2A). This deviation from the ideal theoretical expectation (i.e., no population change during the adiabatic compression and expansion strokes) can be attributed to the fact that in the experiments the frequency of the 729 nm laser was not tuned smoothly in a continuous fashion but instead we used a sequence of discrete steps using an AOM (Supplementary Note 6). When this imperfection is taken into account, we observe a good agreement between theory and our experimental results. So the slight unexpected deviation in the excited state population does not affect our end-results and conclusions on the Otto cycle physics.

Next, we implement both of the isochoric strokes in the broken phase (i.e., the regime with real eigenvalues: non-oscillatory exponentially decaying dynamics). In contrast to the previous case, in which both isochoric strokes were implemented in the exact phase (oscillatory dynamics), the time evolution of the excited state population $P_e$ in this case exhibits neither a hump nor a ramp during the isochoric strokes and $P_e$ stays small during the full Otto cycle (Fig. 2D) due to weak coupling, in agreement with the results obtained by solving the Master equation. This indicates that coherence is largely erased during the isochoric strokes implemented in the broken phase, and thus the net work and output power are much smaller (Fig. 2E). In this regime, we observe that $\eta_c$ stays almost constant and has a value very close to $\eta_O$ as $t_2$ is varied, whereas $\eta_q$ gradually increases and attains a value very close to $\eta_O$ after $t_2 = 4\,\mu s$ (Fig. 2C), indicating that the net work is nearly equivalent to the heat absorbed during the isochoric heating (i.e., 2nd stroke). This is a typical feature of classical Otto cycles, where quantum coherence is not involved.

We note that due to the cold reservoir at the effective temperature $T = 0$ K with $\Delta = 0$, the Otto cycle efficiency $\eta_c$ we observe here is $\eta_c \simeq 99.07\%$, which is much higher than the efficiencies reported previously for heat engines implemented in a $^{13}C$ nucleus[12], a trapped ion[10], and a quantum dot[47]. Moreover, comparison of the different conditions in Figs. 2 and 3 suggests that larger reduction in coherence during an Otto cycle, by executing the isochoric strokes in the broken phase, results in higher $\eta_c$, but significantly reduced net work and output power.

The two sets of experiments discussed above suggest that coherence in the isochoric heating and cooling strokes of the Otto cycle enhances the net work and output power at the expense of the efficiency $\eta_c$, which approaches the ideal value when coherence involved during the isochoric strokes is not high. These experiments also suggest that coherence-enabled heat absorption during the isochoric cooling cycle is the reason for the reduced $\eta_c$, when isochoric strokes are implemented in the exact phase (Supplementary Fig. 4).

These results open a window of opportunity to achieve high $\eta_c$ simultaneously with high net work and output power: Execute the isochoric heating in the exact phase, but the isochoric cooling in the broken phase. Implementing the isochoric cooling in the broken phase will largely reduce the coherence and thus significantly decrease the amount of heat absorption during the cooling stroke. This will prevent

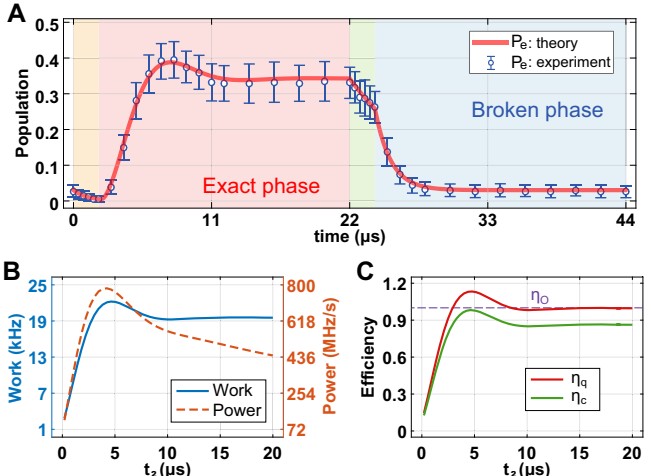

**Fig. 3 | The Otto cycle with the isochoric heating process in the exact phase and the isochoric cooling process in the broken phase. A** Time evolution of the population in the excited state $|e\rangle$ when the second stroke (isochoric heating) is in the exact phase with oscillatory dynamics and the fourth stroke (isochoric cooling) is in the broken phase with non-oscillatory dynamics. The red solid curve is obtained by simulating the master equation and the circles and the error bars respectively denote the average and standard deviation of 10,000 measurements. Regions with different colors correspond to different strokes of the Otto cycle, with the orange, pink, green, and blue corresponding to the first (adiabatic compression), second (isochoric heating), third (adiabatic expansion), and fourth (isochoric cooling) strokes, respectively. The first (orange) and the third (green) strokes are implemented by up- and down-scanning the detuning $\Delta$ between $\Delta_{min}/2\pi = 0$ kHz, and $\Delta_{max}/2\pi = 10$ kHz, respectively, with constant $\Omega$ and $\gamma_{eff}$. The second (pink) and the fourth (blue) strokes are implemented by rapidly increasing and decreasing $\Omega/\gamma_{eff}$ with constant detuning. We used $\{\Omega/2\pi = 25\,\text{kHz}, \gamma_{eff} = 860\,\text{kHz}\}$, and $\{\Omega/2\pi = 25\,\text{kHz}, \gamma_{eff} = 140\,\text{kHz}\}$ for the first and third strokes, respectively, and $\{\Omega/2\pi = 90\,\text{kHz}, \gamma_{eff} = 500\,\text{kHz}, \Delta = \Delta_{max}\}$ and $\{\Omega/2\pi = 25\,\text{kHz}, \gamma_{eff} = 860\,\text{kHz}, \Delta = \Delta_{min}\}$ for the second and fourth strokes, respectively. **B** The net work and output power, and (**C**) the efficiencies $\eta_c$ and $\eta_q$ as functions of the second stroke execution time $t_2$ ($\leq 20\,\mu s$), while the execution times of the other strokes are kept fixed. In (**C**), the horizontal dashed line represents the efficiency $\eta_O = 1$ of the ideal Otto cycle.

the reduction in $\eta_c$ and at the same time generates more work and power. In order to confirm this prediction, we have designed the final experiments in such a way that the isochoric heating is executed in the exact phase, whereas the isochoric cooling is executed in the broken phase. The time evolution of the excited state population $P_e$ in this case exhibits a hump during the heating stroke which is executed in the exact phase and no ramp is seen during the cooling stroke, which is executed in the broken phase (Fig. 3A). This implies the existence of sufficient coherence only during the heating stroke but not during the cooling stroke. The maximum value of $P_e$ is as high as the value obtained for the case when both isochoric strokes are executed in the exact phase. The hump (which is the signature of the presence of sufficient coherence) is seen in the plots of the net work, output power, $\eta_q$ versus $t_2$, and $\eta_c$ versus $t_2$ (Fig. 3B, C). The net work and the output power obtained in this setting (Fig. 3B) are much higher than the previous two settings, whose results are given in Fig. 2B, E. Moreover, we obtain $\eta_c \approx 1$ for $t_2 \leq 5\,\mu s$, implying that the net work is nearly equivalent to the absorbed heat during the heating stroke. This is the result of the absence of sufficient coherence and hence less heat absorption during the cooling stroke.

Note that $\eta_c$ experiences only a very small decrease down to $\eta_c \approx 0.9$ for longer $t_2$, which is much higher than the $\eta_c$ - 0.65 obtained when both isochoric strokes are executed in the exact phase, and only slightly lower than the $\eta_c = 1$ obtained when both strokes are executed in the broken phase. We note that the net work, output power, $\eta_c$, and $\eta_q$ achieved their maximum at $t_2 = 4\,\mu s$. Our experiments clearly

demonstrate that the exact phase (and hence more coherence) during the isochoric heating (second stroke) and broken phase (and hence less coherence) during the isochoric cooling (fourth stroke) lead to a better quantum heat engine performance.

## Discussion

We have implemented a quantum heat engine using a single trapped ion and elucidated the role of coherence, LEPs, and the associated dynamics in the performance of a heat engine. This was done by executing the isochoric heating and cooling strokes of an Otto cycle, respectively, in the exact- and broken-phases separated in the parameter space by an LEP. Our experimental observations are fully understood by the Lindblad master equation, which takes quantum jumps into account, and thus LEP represents a proper description of purely quantum effects involved in an Otto cycle. We have shown that the highest net work, output power, and efficiency can be achieved when the isochoric heating and cooling strokes are executed respectively in the exact phase (oscillatory dynamics and higher coherence) and broken phase (exponentially decaying dynamics and no or less coherence). This is in contrast to the conventional view that coherence in the isochoric strokes is crucial for enhanced performance of a quantum engine. This counterintuitive result would help understand thermodynamic effects in non-Hermitian systems exhibiting EPs and the role of quantum effects in heat-work conversion and working substance-bath interaction in heat engines. Our results open interesting possibilities for the control of the dynamics of quantum heat engines and will be of interest to other research areas that are concerned with the role of coherence and EPs in the enhancement of quantum processes.

Further study would evolve in two directions. First, one of the vibrational modes of the trapped ion working as the heat engine can be used as the quantum load (i.e., optical states of the ion act as the working substance and the vibrational modes coupled to them act as the load) and heating and cooling processes can be studied in the spirit of sideband heating or sideband cooling. Second, an additional ion confined in the same trap with the ion working as the heat engine can be used as the load. One can then rearrange the strokes of the engine cycle to perform heating or refrigeration. For example, performing the strokes of the Otto cycle in counterclockwise direction as shown in Fig. 1E with the stroke sequence as $1 \to 2 \to 3 \to 4 \to 1$ will lead to heating whereas carrying out the Otto cycle in the clockwise direction with the stroke sequence as $1 \to 4 \to 3 \to 2 \to 1$ will result in cooling. In these cases, the engine will be coupled to (decoupled from) the load during the adiabatic compression and expansion strokes (isochoric heating and cooling strokes). Performing measurements on the load after each engine cycle would then help understanding the cooling and heating process as a function of the number of engine cycles. One should however keep in mind that correlations may build up between the quantum engine and the quantum load during the adiabatic strokes (when they are coupled); therefore, one should be careful when interpreting heating/cooling, work, and other thermodynamic quantities. Further studies are needed to have a deeper physical insight into the role of LEPs in the performance of quantum heat engines and to better quantify the heat, power, and efficiency of quantum heat engines coupled to quantum loads.

## Data availability

The data illustrated in the figures within this paper are available from the corresponding authors upon request. The data are also available at https://www.scidb.cn/en/anonymous/eTZ2QTNx.

## Code availability

The custom codes used to generate the results presented in this paper are available from the corresponding authors upon request.

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

## Acknowledgements

The authors are grateful to Yunlan Zuo for her help in plotting Fig. 1. This work was supported by Special Project for Research andDevelopment in Key Areas of Guangdong Province under Grant No. 2020B0303300001, by National Natural Science Foundation of China under Grant Nos. U21A20434, 12074346, 12074390, 11835011, 11804375, 11804308, 91636220, by Postdoctoral Science Foundation of China under Grant No. 2022MT10881, by Natural Science Foundation of Henan Province under Grant No. 202300410481 and 212300410085, by K.C. Wong Education Foundation (GJTD-2019-15). S.K.O acknowledges the support from Air Force Office of Scientific Research (AFOSR) Multidisciplinary University Research Initiative (MURI) Award on Programmable Systems with Non-Hermitian Quantum Dynamics (Award No. FA9550-21-1-0202). F.N. is supported in part by Nippon Telegraph and Telephone Corporation (NTT) Research, the Japan Science and Technology Agency (JST) [via the Quantum Leap Flagship Program (Q-LEAP), and the Moonshot R&D Grant Number JPMJMS2061], the Japan Society for the Promotion of Science (JSPS) [via the Grants-in-Aid for Scientific Research (KAKENHI) Grant No. JP20H00134], the Army Research Office (ARO) (Grant No. W911NF-18-1-0358), the Asian Office of Aerospace Research and Development (AOARD) (via Grant No. FA2386-20-1-4069), and the Foundational Questions Institute Fund (FQXi) via Grant No. FQXi-IAF19-06.

## Author contributions

H.J. and M.F. conceived the idea and designed the experiments; J.W.Z. and F.Z. performed the experiments with help from J.C.L., J.T.B., and B.W.; J.Q.Z., D.G.Y., L.L.Y., S.L.S., and L.C. analyzed the data. Theoretical background and simulations were provided by J.Q.Z. and G.Y.D. All authors contributed to the discussions and the interpretations of the experimental and theoretical results. M.F., H.J., F.N., and S.K.O. wrote the paper with inputs from all authors.

## Competing interests

The authors declare no competing interests.
