## [Peer Review File · Nature Communications]

Dynamical Control of Quantum Heat Engines Using Exceptional PointsREVIEWER COMMENTS

Reviewer #1 (Remarks to the Author):

The authors present a paper in which they claim to realize a quantum thermal machine with a single trapped ion. To do so, they study the population dynamics of a three level system which is driven by two lasers. One of them can generate coherent via Rabi oscillations on a qubit defined between two of the three levels and another one generates dissipation by induced spontaneous emission via the short lived state. By choosing different values for the laser powers and frequencies they emulate adiabatic and isochoric processes on the two level system. They then calculate quantities which they associate with work, heat and efficiency. They also solve a model for the dynamics of the system and find a good agreement with the observed data.

I believe the claim that they “operating” or “realizing” a quantum heat engine is incorrect. At the most, what they do is emulate a part of a heat engine. As so, it is my opinion that these results are, as presented, not valid for their publication in their current form. Moreover, if the overstatements were toned down and made reflect what it is actually being done, I believe they will not be worthy for their publication in Nature Communications, but would be perfectly suitable for another, lower impact journal. I do believe the experimental and theoretical work can be taken to a reasonable stand, so they could be published, with an important rework of the wording and the overall interpretation.

I base my judgment on the following observations:

First and foremost. Engines consist of a working mechanism and a dynamical system on which the work is done or to which it is delivered. This seems to be absent in the current presentation. Here, there is only the qubit, which as I understand, can be interpreted only as the working medium. It is unclear to me, if anywhere, the calculated work can be accumulated or extracted, even could potentially be accumulated or extracted in the presented system.

Also related to this, is the fact that the authors only study one period of the allegedly thermal engine. It seems clear to me, from the data and their statements, that more periods would give the same result. Then one could ask oneself, is it because the engine has reached equilibrium between the work it does and the dissipation in the load, or is it because no power is delivered to no load. I understand the answer is the latter, and as such this is no thermal machine, but at the most, an emulation of one part of it.

On another important point, the fact that an increase in the laser detuning of the laser can simulate or mimic an adiabatic compression is not clear at all. It should be discussed with more detail. For a harmonic oscillator, an adiabatic compression and expansion have long been understood as increasing or decreasing the oscillator frequency, thereby changing the energy spacing. Instead, here the authors, claim they realize adiabatic processes by tuning the coherent laser into and out of resonance. How this could represent the former situation is at least not clear from the text, and somewhat unclear to me.

Finally, in page three the authors discuss the efficiency of the thermal engine. They calculate this efficiency to be above 99% and compare to other realizations of quantum heat engines, which reported lower values. They then suggest this increase is due to the reduction of coherence in the cycle for the dissipative regime. To this I must say the following. First, it is no wonder they have very high efficiency, as one of the heat baths is operated at almost absolute zero temperature. The authors are aware of that fact, as they start the paragraph explicitly saying that, but then conclude otherwise. If there were an increase in efficiency due to the presence or not of coherences, this fact should be proved otherwise. For example, by comparing efficiencies of different cycles, with equivalent Carnot efficiency, or by any means that is actually proving or at least suggesting this fact.

On the bright side of the paper, I must say I liked the discussion about the two regimes the authors called broken- and exact-phase. The dynamics and the physics is clear and the data shows clearly

that in one case one has pure dissipation-dominated dynamics while in the other, coherence starts to kick in showing oscillations. Though these regimes and behavior is well know, I find it a nice presentation and interesting results. However I must say, that I found that the excessive pushing of the jargon “Liuvillian Exceptional Points”, “Exact-” and “Broken-phases”, seem to me to come as a whole as a bit of overselling a phenomenon that is already well known.

As said, I would suggest the authors to review their interpretations and naming of what they are doing, and consider submitting to another journal after that.

Some other comments to make the manuscript better:

They author state “This deviation (...) can be attributed to the imperfect frequency modulation of the 729-nm laser.” Here, it is not clear to me by imperfect frequency modulation. It should be cleared up. Also, next, they claim that the population changes during the adiabatic strokes “do not affect the physics of the Otto cycle”. Why not? I think this should be explained since it does not seem obvious.

Check the sentence starting with “The strong DTD regime and the exact phase overlap for” .. I do not understand it, there seems to be a part missing or some grammar issue.

Reviewer #2 (Remarks to the Author):

Dear authors, dear Editor,

As mentioned to the Editor, in view of a fully transparent review process, I was asked to refer your paper when submitted earlier to another journal. The Editor asked me to also be referee for your paper for your submission to Nature Communications and allowed me to explicitly build this report on my previous one.

For the Editing review process, I will copy and paste below the main questions / comments I made during my last review round. The authors have modified their manuscript in a very serious and rigorous way, taking into account most of the comments I made, especially with respect to recent literature on the topic of LEPs. To my opinion, the role of quantum coherence has still to be understood in deep details (see my comments below) but the authors have moderated their statements compared to the earlier version. In the present form, the manuscript raised very interesting physics that will trigger interest in the community of people interested in the dynamics of open quantum systems.

In summary, the objective and main questions raised by this work are very timely and of interest for a broad scientific community (open quantum systems, non-Hermitian physics, quantum thermodynamics, quantum control to cite some). It addresses fundamental questions and the approach and designed experiment are well suited to pursue these objectives. The manuscript is well-written, with a clear presentation of the results. I think it meets all the standards of Nature Communications and I support its publication.

With my best regards.

%%%

Comments / questions from earlier report

1) The authors write that LEPs correspond to the transition points between the exact and broken PT phases. This is not correct. LEPs do not necessarily correspond to that in the sense that you can find LEPs (and HEPs) for non Parity-Time symmetric systems. It is true that for a PT-symmetric system, LEPs do correspond to the transition between exact and broken phases, but the reverse is not always true. In this manuscript, the Parity Time operators are not defined, and it is not checked that this

system is PT symmetric. This is missing. If the model is shown to be PT-symmetric, then the authors can write that LEPs will correspond to the transition, and this will be a rigorous statement.

2) The authors put forward an analysis of their results where the presence or absence of coherence would be related to being in the broken and exact PT-phases, see discussion page 10 and after. To my understanding, and in view of Khandelwal et al., PRX Quantum 2 (theory paper) and Chen et al., arXiv:2111.04754 (2021) (experimental paper), I would tend to put forward another explanation, related to LEP being the point of critical damping of the system, i.e. the limit point in parameter space between oscillatory and non-oscillatory dynamics, in analogy with the damped harmonic oscillator.

Indeed, in the exact phase (strong driving), eigenvalues λ_3 and λ_4 have an imaginary component, inducing an oscillatory behavior, which is fully compatible with the hump and ramp observed in Fig. 2A1, 2A2 and 2A3. In contrast, in the broken phase (weak driving), eigenvalues λ_3 and λ_4 are real, inducing a purely exponential decaying dynamics, i.e. non-oscillatory dynamics, fully compatible with Fig. 2B1, 2B2, and 2B3.

One can also note that the same oscillatory versus non-oscillatory behavior seems to make sense also in view of Fig. S3, showing the norm of the coherence as a function of time.

Can the authors comment? I strongly believe that this is a very nice experiment and very timely topic, and it would be very desirable for the community to see whether one can make sense of the latest experiments and theory works. Of course, this may require additional work from the authors, but this may be worth it. I am very interested in the authors' response.

3) One can also question the interpretation of the role of coherence in the following way. From Fig. S3, it seems that the order of magnitude of the norm of coherence is similar in the broken and exact PT-phase (smaller in the broken PT phase, but clearly not negligible as compared to the norm in the exact phase). Hence, when the authors write page 10 that coherence is completely erased in the isochoric strokes implemented in the broken PT-phase, I do not agree. Can the authors comment / explain? This observation led me to consider another explanation for the dynamical behavior, hence my above comment, see 2).

(Minor: when the authors make this statement, there is a typo in isochoric.)

4) The authors mention briefly the bifurcation seen in Fig. 1D. The origin of this bifurcation remains unclear for me after reading the manuscript. Could the authors expand on this?

5) In Fig. 2, the error bars seem to be (much) larger in the broken PT phase than in the exact PT phase. Is there a reason behind?

6) Comment about the formulation page 8, second paragraph: The authors start the second paragraph by "Interestingly. The strong DTD regime and exact PT phase overlap for $\Omega/g_{\text{eff}} > 174$ [...]". I find the adverb "interestingly" surprising. Indeed, the strength of the DTD depends on Ω versus g_{eff} (by definition), and those two parameters are exactly the ones allowing for controlling the term under the square root in the eigenvalues λ_3 and λ_4 . So, the "overlap" does not happen by chance, it's more by definition of their DTD, that the exact phase is reached in the strong DTD regime and vice-versa.

7) Page 9, the authors discuss Fig. 2A3, and in statement iii), claim that η_c is always smaller than η_q . From the figure, this statement does not seem to be true at short times. Can the authors explain what they mean then?

8) About references, there are clearly two important ones missing, cited earlier in this report:

- Khandelwal et al., PRX Quantum 2, 040346 (2021)

- Chen et al., arXiv:2111.04754 (2021)

As it is a very attractive research direction right now, the authors should also double-check the very

recent literature before resubmission.

Reviewer #3 (Remarks to the Author):

In this paper, the authors study a quantum heat engine that features so-called Liouvillian exceptional points. They find that when the heat engine operates in the exact phase during isochoric heating, while operating in the broken phase during cooling strokes of an Otto cycle results in more work and output power, while similarly leading to higher efficiency as compared to performing the Otto cycle in the exact phase.

With this work, the authors demonstrate the implications of the presence of a Liouvillian exceptional point. As the authors point out, typically, so-called Hamiltonian exceptional points are studied and there is not much work on studying the consequences of Liouvillian exceptional points. As such, this paper presents an interesting addition to the expanding field of non-Hermitian topology.

The paper is well written, and the figures complement the text. As such, I recommend this work for publication. I nevertheless have one comment for the authors. While the authors point out the importance of parity-time symmetry in their work, they nowhere actually address what symmetry is present in their model. As they are referring to exact and broken phases with relation to the Liouvillian exceptional point, they are borrowing language from the PT community. Could the authors comment on what symmetry their system has?

List of the Main Changes

1. We have rewritten the first sentence of the second paragraph on page 8 of the revised manuscript to explain our operations for quantum adiabatic stroke. *[for Reviewer 1 Comment 3]*
2. We have rewritten the sentence in the third paragraph on page 11 of the revised manuscript to mention the relation of the quantum coherence to the efficiency. *[for Reviewer 1 Comment 4]*
3. We have rewritten the sentence in the last paragraph on page 9 of the revised manuscript to explain the imperfect frequency modulation in our experiment. To further clarify this point, we have added a paragraph and a supplementary figure in Supplementary material. *[for Reviewer 1 Comment 5]*

To help the editor and the Reviewers better understand our revision, we have highlighted the main revision in the pdf version of the submitted files.

RESPONSE TO REVIEWER # 1

We appreciate the Reviewer's critical reading of our manuscript and the specific comments. To clarify some key points, we respond to the comments one-by-one as below. We believe that our revised manuscript addresses the issues raised by the Reviewer.

Comment 1:

Engines consist of a working mechanism and a dynamical system on which the work is done or to which it is delivered. This seems to be absent in the current presentation. Here, there is only the qubit, which as I understand, can be interpreted only as the working medium. It is unclear to me, if anywhere, the calculated work can be accumulated or extracted, even could potentially be accumulated or extracted in the presented system.

Our Response 1:

We agree with the Reviewer that a working engine should demonstrate the work to be done and delivered. However, we regrettably do not agree with the Reviewer that what we have carried out experimentally is just for a qubit.

As mentioned in our manuscript, the quantum heat engine designed and implemented in our experiment includes the working medium (i.e., the qubit encoded in the ion) and the baths (i.e., the laser irradiation together with the real environment with the hot and cold baths defined by strong and weak laser drives, respectively). We build a link between the experimentally observable population of the qubit and the work (and heat) accumulated or delivered by the quantum heat engine (see Supplementary Note 3). Since the real environment is part of the baths in our model, the work could be delivered by dissipation to the environment. We describe our open system by the master equation, which has traced over the environment. Thus, by monitoring the population of the qubit, we know how much work is delivered to the environment. Similarly, the strong laser drive, as a hot bath, performs work on the system, which can also be acquired quantitatively by observing the variation of the population. Consequently, this is a complete model of the heat engine, which demonstrates the fundamental mechanism of the heat engine at a single-qubit level. By this way,

we have revealed in our experiment under which condition the higher net work, output power, and efficiency can be achieved in Otto cycles.

Comment 2:

the authors only study one period of the allegedly thermal engine. It seems clear to me, from the data and their statements, that more periods would give the same result. Then one could as oneself, is it because the engine has reach equilibrium between the work it does and the dissipation in the load, or is it because no power is delivered to no load. I understand the answer is the latter, and as such this is no thermal machine, but at the most, an emulation of one part of it.

Our Response 2:

To demonstrate the nontrivial features of the single-qubit quantum heat engine, we focus our study on a close cycle of the Otto engine. To this end, after finishing the last stroke, we wait for the system to reach the steady state and then return to the initial state. It is true that our system finally reaches an equilibrium state after accomplishing one cycle.

But this does not mean that no power is delivered in the cycle. As we mentioned in our response to comment 1, the work of the quantum heat engine is delivered by dissipation to the environment. In fact, if necessary, we may repeatedly perform the engine cycles, delivering more power to the baths, i.e., the environment. Since our focus in this study has been to clarify how the Liouvillian exceptional points (LEPs) affect the heat engine dynamics and performance, it is sufficient to perform only one Otto cycle and then wait until the system reaches the equilibrium. We could very well perform multiple cycles without waiting the system reach equilibrium state. This is possible in our system and is not a limitation. We think the completed one cycle is enough to answer the questions motivated this research.

Comment 3:

an increase in the laser detuning of the laser can simulate or mimic an adiabatic compression is not clear at all. It should be discussed with more detail. For a harmonic oscillator, an adiabatic compression and expansion have long been understood as increasing or decreasing the oscillator frequency, thereby changing the energy spacing. Instead, here the authors, claim they realize adiabatic processes by tuning the coherent laser into and out of resonance. How this could represent the former situation is at least not clear from the text, and somewhat unclear to me.

Our Response 3:

We agree with the Reviewer that increasing or decreasing the oscillator frequency (or the energy spacing) accomplishes the adiabatic compression and expansion, implying that the entropy remains unchanged. In our model, the qubit system is manipulated by the lasers, changing the detuning Δ and the coupling strength Ω , as described in the interacting representation (i.e., a rotating frame with respect to the laser frequency) by Supplementary Eq. (9), and the entropy is defined by the populations of the two levels. Therefore, in this interacting representation, changing the detuning (i.e., the difference between the laser frequency and the two-level resonance frequency) actually corresponds to the variation of the energy spacing without the population change, corresponding to an isentropic process (i.e., a thermodynamic process that is both adiabatic and reversible). To clarify

this point, we have rewritten the following sentence in the second paragraph on page 8 of the revised version of the manuscript:

“In our treatment under the rotating frame with respect to the driving laser frequency, quantum adiabatic strokes are executed by tuning the frequency of the driving laser which helps vary the internal energy gap Δ (but without population change) and the temperature of the working substance.” --

This above modification, together with the sentence “In a quantum adiabatic stroke, P_n and S should remain unchanged during the process (thus no heat exchange) but E_n may shift” in the second paragraph on page 7 and the definitions in Supplementary Note 3 should fully clarify the question raised by the Reviewer.

Comment 4:

in page three the authors discuss the efficiency of the thermal engine. They calculate this efficiency to be above 99% and compare to other realizations of quantum heat engines, which reported lower values. They then suggest this increase is due to the reduction of coherence in the cycle for the dissipative regime. To this I must say the following. First, it is no wonder they have very high efficiency, as one of the heat baths is operated at almost absolute zero temperature. The authors are aware of that fact, as they start the paragraph explicitly saying that, but then conclude otherwise. If there were an increase in efficiency due to the presence or not of coherences, this fact should be proved otherwise. For example, by comparing efficiencies of different cycles, with equivalent Carnot efficiency, or by any means that is actually proving or at least suggesting this fact.

Our Response 4:

We agree with the Reviewer that the bath temperature approaching zero could lead to the high efficiency of our engine cycle. But our experimental observation in the single-spin quantum heat engine also presents the link between the coherence and the efficiency. We have performed the experiments under three conditions, in each of which $\Delta_{\min}=0$ kHz is reached at the end of the fourth stroke, implying zero temperature in the system-bath equilibrium state. However, we can find the efficiency difference in the three cases, see Figs. 2A₃, 2B₃ and 3C. The higher efficiency appears in the cases involving the broken phase, i.e., with reduced coherence. As such, we attribute the higher efficiency to the reduced coherence. To further clarify this point, we have rewritten the relevant sentence in the third paragraph on page 11 of the revised manuscript as follows:

“Moreover, comparison of the different conditions in Figs. 2 and 3 suggests that larger reduction in coherence during an Otto cycle, by executing the isochoric strokes in the broken phase, results in higher η_c , but significantly reduced net work and output power.”

This modification, together with discussions about the coherence-enabled performance in the subsequent paragraphs should fully clarify the point.

Comment 5:

They author state “This deviation (...) can be attributed to the imperfect frequency modulation of the 729-nm laser.” Here, it is not clear to me by imperfect frequency modulation. It should be cleared up. Also, next, they claim that the population changes during the adiabatic strokes “do not affect the

physics of the Otto cycle". Why not? I think this should be explained since it does not seem obvious.

Our Response 5:

Theoretically, we need to tune the frequency of 729-nm laser continuously in time to accomplish the first and third strokes. Experimentally, however, this frequency tuning is carried out discretely by tuning the acousto-optic modulator in sequence, which would bring in unexpected phases (see the Figure below), leading to slight population drops (~ 0.1) in the two strokes, as observation in, e.g., Fig. 2A₁. Considering these unexpected phases in numerical simulation, we have fitted the experimental observations involving such cases by theory very well, see Figs. 2A₁, 2B₁ and 3A. Moreover, to avoid misunderstandings, we have rewritten the relevant sentences for this point in the last paragraph on Page 9 of the revised version of the manuscript as follows:

"This deviation from the ideal theoretical expectation (i.e., no population change during the adiabatic compression and expansion strokes) can be attributed to the fact that in the experiments the frequency of the 729 nm laser was not tuned smoothly in a continuous fashion but instead we used a sequence of discrete steps using an AOM."

The detuning Δ should be varied continuously as in the Blue line in the figure. However, in our experiments we approximate such tuning of Δ as a sequence of discrete steps using an acousto-optic modulator (Red line). This results in a tuning similar to that shown in the Green curve, which is a deviation from the ideal tuning line and thus brings in unexpected phases in the operation of the system. This deviation from the ideal tuning curve is what we referred to as imperfect frequency modulation.

We have also added the above figure and the explanation in the Supplement to make it clear and cite it in the main text.

Comment 6:

Check the sentence starting with "The strong DTD regime and the exact phase overlap for" .. I do not understand it, there seems to be a part missing or some grammar issue.

Our Response 6:

We thank the Reviewer for bringing this to our attention. This was something left in the manuscript during our many rounds of revisions. Since we have defined “strong coupling” (“weak coupling”) in the original version of the manuscript, which fully overlaps with the exact-phase (broken-phase) regime, the definition of “DTD” is unnecessary. We have replaced “strong (weak) DTD” by “strong coupling” (“weak coupling”) in the revised version of our manuscript.

RESPONSE TO REVIEWER # 2

We thank the Reviewer for being transparent in the review process, and we also thank him for very constructive and useful comments in their previous report. The comments made us become aware of the literature and the various ways of interpreting our experimental results. We are happy to hear that the Reviewer has found our revisions in the current manuscript to be sufficient and rigorous. We agree with the Reviewer that the role of coherence in the quantum heat engine and its connection to exceptional point physics need further studies and detailed analysis which we hope to address and research in the future. Finally, we want to thank the Reviewer for finding our study to be timely, acknowledging its importance, and recommending it for publication.

RESPONSE TO REVIEWER # 3

We thank Reviewer #3 for his/her serious reading of our paper, for acknowledging its importance, and recommending it for publication. Below we address the Reviewer’s only question in their new report.

Comment 1:

As they are referring to exact and broken phases with relation to the Liouvillian exceptional point, they are borrowing language from the PT community. Could the authors comment on what symmetry their system has?

Our Response:

We agree with the Reviewer that we “are borrowing the language from the PT community”, in the sense that exact PT phase corresponds to formation of supermodes and Rabi-like oscillatory behavior with real eigenvalues; the broken PT phase corresponds to coalescence of modes in frequency with damping/amplifying behavior; and the exceptional point marking the transition between these two regimes. In our experiments and theoretical consideration, we observe the same behavior and LEP marks the transition between oscillatory and damped behavior. That is why we borrowed the language from the PT community. However, we should note that we have not yet found the parity operator for the Hamiltonian when the quantum jump is involved. Instead, we present the eigensolution details of the Liouvillian superoperator in Supplementary Note 2.

REVIEWERS' COMMENTS

Reviewer #1 (Remarks to the Author):

Dear Authors.

Thanks for the detailed response to my points raised.

I am satisfied with the answers concerning technical issues that were not clear to me in the previous version. However, I must say that I am not convinced of the explanations presented to justify my main criticisms on the interpretation of the results, as I explain below.

First, to my objection that the heat engine is lacking a medium on which the work is done, the authors explain the work is done to the environment. Though formally a correct answer, this sounds like an oxymoron to me. It breaks the whole idea of doing work onto something.

Moreover, in response to my objection that they only look at one cycle of this device, I see new evidence of this futile machine: as it does the work on an effectively infinite environment, each cycle is equal to the next.

All that said, I think, as referee 2 properly states, "the role of quantum coherence (in thermodynamic processes) has still to be understood in deep details". In this respect, this paper shows an interesting phenomena. It shows that a machine working part of its time in two regimes, one dominated by decoherence and another with some coherence might lead to better results than if it's only working on one of them. Combining both high efficiency (as in classical heat engines) and high output power (in their version with coherences).

Finally I would like to add a new observation I came upon a new reading of the manuscript. On one hand, I noticed that the authors decided to operate with "long" second strokes. This means they actually lose part of the benefit of the coherence. More specifically, they choose times for t_2 that are longer than the coherence humps. If they would choose t_2 to match the coherence hump, the performance of the machine working all in the coherence regime (exact phase) would be very similar to that of the mixed regime (exact and broken phases).

The above observation leads me to ask if this process could be interpreted as a motional cooling process, as a refrigerator, instead of a heat engine. Actually pulsed sideband cooling works in such a fashion, but where the detuning matches a sideband. If such an interpretation were possible, it would solve the conceptual points raised above.

All that said, seeing the other referees comments and given this is an open review process, I would not be against this article being published in Nature Communications, given this discussion will be printed alongside. I believe, opening these issues to the community might help the understanding and development of a field where there are still fundamental issues left to understand.

List of the Main Changes

1. We have deleted the words such as new/novel/first in the main text and the supplemental material, following the editorial policy.
2. We have added some sentences in the part of Discussion as the last paragraph of the main text for responding to the Reviewer's new comments. This new paragraph is highlighted in the pdf file "NV-Main-Resub2-highlight.pdf".

RESPONSE TO REVIEWER # 1

We thank the Reviewer for the constructive comments and their statement "I would not be against this article being published in Nature Communications". We agree with the Reviewer that our discussion and arguments presented in this manuscript "might help the understanding and development of a field where there are still fundamental issues left to understand." In the following we provide our responses to the Reviewer's comments. We believe that our revised manuscript addresses the issues raised by the Reviewer.

Comment 1:

First, to my objection that the heat engine is lacking a medium on which the work is done, the authors explain the work is done to the environment. Though formally a correct answer, this sounds like an oxymoron to me. It breaks the whole idea of doing work onto something.

Our Response 1:

We understand what the Reviewer means, and we respectfully do not agree with the Reviewer that our heat engine "breaks the whole idea of doing work onto something". We would like to mention again that what we designed is a complete model of the heat engine and our experiment demonstrates the fundamental mechanism of the heat engine at a single-qubit level. By this way, we have revealed experimentally under which condition the higher net work, output power, and efficiency can be achieved in Otto cycles.

Comment 2:

Moreover, in response to my objection that they only look at one cycle of this device, I see new evidence of this futile machine: as it does the work on an effectively infinite environment, each cycle is equal to the next.

Our Response 2:

We have responded to a similar comment from this Reviewer in the last review round. Here we would like to mention again that the motivation of our work is to clarify how the Liouvillian exceptional points affect the heat engine dynamics and performance. As such, one close cycle of the Otto engine is sufficient to show the important roles of quantum coherence and the Liouvillian exceptional points played in such a single-qubit quantum heat engine. The Reviewer is correct in their assessment that since the engine in this study is connected to an infinite environment, each cycle is equal to the next cycle. This will not be the case if a quantum load is connected to the engine. To address this point, we have added sentences in the discussion part of the manuscript. The added paragraph is listed below in our response to the 4th comment of the Reviewer.

Comment 3:

All that said, I think, as referee 2 properly states, "the role of quantum coherence (in thermodynamic processes) has still to be understood in deep details". In this respect, this paper shows an interesting phenomena. It shows that a machine working part of its time in two regimes, one dominated by decoherence and another with some coherence might lead to better results than if it's only working on one of them. Combining both high efficiency (as in classical heat engines) and high output power (in their version with coherences).

Our Response 3:

The Reviewer here provides a nice summary of what we have intended to achieve and demonstrate in this work. We thank the Reviewer for finding our results on the role of coherence in heat engines to be interesting.

Comment 4:

Finally I would like to add a new observation I came upon a new reading of the manuscript. On one hand, I noticed that the authors decided to operate with "long" second strokes. This means they actually lose part of the benefit of the coherence. More specifically, they choose times for t_2 that are longer than the coherence humps. If they would choose t_2 to match the coherence hump, the performance of the machine working all in the coherence regime (exact phase) would be very similar to that of the mixed regime (exact and broken phases). The above observation leads me to ask if this process could be interpreted as a motional cooling process, as a refrigerator, instead of a heat engine. Actually pulsed sideband cooling works in such a fashion, but where the detuning matches a sideband. If such an interpretation were possible, it would solve the conceptual points raised above.

Our Response 4:

Since our work is motivated to investigate the role of quantum coherence in the work output and efficiency of the heat engine, we have performed experiments to observe long time evolutions in the isochoric strokes. Our experimental and numerical simulation results reveal that the highest net work performed, output power, and efficiency can be achieved when the isochoric heating and cooling strokes of the heat engine are executed respectively in the exact and broken phases.

The Reviewer, as we understand from the comment, wants to know whether our results and conclusions will change if the t_2 time of second stroke is comparable or shorter than the coherence times. As shown in Figs. 2A₂, 2A₃, 2B₂, 2B₃, 3B and 3C, we have depicted the net work performed, output power, and efficiency of the heat engine as a function of t_2 . These results show that regardless of whether both strokes are in the exact phase (Fig. 2A), in the broken phase (Fig. 2B), or isochoric heating and cooling are respectively in the exact and broken phases (Fig. 3), the net work done, power, and the efficiency of the heat engine reach their maximum at t_2 matching the time scale of coherence humps. This is reasonable due to the higher quantum coherence at t_2 times matching the time scale of coherence humps. A comparison of these three cases, on the other hand, reveals that the highest net work, power, and the efficiency values are obtained for the case when the isochoric heating and cooling are respectively in the exact and broken phases and the t_2 time is matched to the time of the coherence hump. Thus, the scenario mentioned in the Reviewer's comment is not possible, that is the "mixed case" will always result in better work, power, and efficiency.

Since the net work of the engine cycle demonstrated here is proportional to the population difference between the isochoric heating and cooling strokes (See Supplementary Note 3), one may wonder if there exists a time t_2 in the Otto cycle executed in counterclockwise direction (as shown in Fig. 1E with the stroke sequence as $1 \rightarrow 2 \rightarrow 3 \rightarrow 4 \rightarrow 1$) when the population at the fourth stroke is higher than the population at the second stroke and thus the engine works as a quantum refrigerator. In order to address this issue, we have performed numerical simulations which revealed that even at very short t_2 times, the population in the fourth stroke is always lower than that in the second stroke, implying no possibility as a quantum refrigerator in the Otto cycle executed in counterclockwise direction with the stroke sequence as $1 \rightarrow 2 \rightarrow 3 \rightarrow 4 \rightarrow 1$ (see Fig. R1).

Figure R1. Population difference between the second and fourth strokes as a function of time t_2 when both of the isochoric strokes are set in the exact phase. To compare with the results shown in Fig. 2A1, in the simulations we kept the time durations of the other three strokes unchanged and shorten t_2 of the second stroke. Clearly, the population difference is always positive, implying that the engine works as a heat engine rather than a refrigerator even when t_2 time is set very short.

An Otto engine would work as a refrigerators when the cycle is performed in the clockwise direction with the stroke sequence as $1 \rightarrow 4 \rightarrow 3 \rightarrow 2 \rightarrow 1$ as labelled in Fig. 1E or when the isochoric heating stroke is performed in the broken phase but isochoric cooling stroke is performed in the exact phase, as discussed on page 7 of the main text. In these cases, our Otto engine would work as a quantum refrigerator, totally reversing the process plotted in Fig. 3. We thank the Reviewer for bringing the cooling and refrigeration and possibilities along these directions to our discussions. We find the idea of cooling motional degrees of freedom in our system and studying the effects of LEP degeneracies on the cooling efficiency as interesting to pursue experimentally in the future. We have added the following text as the last paragraph of the manuscript:

Further study would evolve in two directions. First, one of the vibrational modes of the trapped ion working as the heat engine can be used as the quantum load (i.e., optical states of the ion act as the working substance and the vibrational modes coupled to them act as the load) and study heating and cooling process in the spirit of sideband heating or sideband cooling. Second, an additional ion confined in the same trap with the ion working as the heat engine can be used as the load. One can then rearrange the strokes of the engine cycle to perform heating or refrigeration. For example, performing the strokes of the Otto cycle in counterclockwise direction as shown in Fig. 1E with the stroke sequence as $1 \rightarrow 2 \rightarrow 3 \rightarrow 4 \rightarrow 1$ will lead to heating whereas carrying out the Otto cycle in the clockwise direction with the stroke sequence as $1 \rightarrow 4 \rightarrow 3 \rightarrow 2 \rightarrow 1$ will result in cooling. In these cases, the engine will be coupled to (decoupled from) the load during the adiabatic compression and expansion strokes (isochoric heating and cooling strokes). Performing measurements on the load after each engine cycle would then help understanding the cooling and heating process as a function of the number of engine cycles. One should however keep in mind that correlations may build up between the quantum engine and the quantum load during the adiabatic strokes

(when they are coupled); therefore, one should be careful when interpreting heating/cooling, work, and other thermodynamic quantities. Further studies are needed to have a deeper physical insight into the role of Liouvillian exceptional points in the performance of quantum heat engines and to better quantify the heat, power, and efficiency of quantum heat engines coupled to quantum loads.